# Multimorbidity in the Elderly: A Systematic Bibliometric Analysis of Research Output

**DOI:** 10.3390/ijerph19010353

**Published:** 2021-12-30

**Authors:** Xuan Zhou, Dan Zhang

**Affiliations:** Institute for Hospital Management, Tsinghua Shenzhen International Graduate School, Tsinghua University, Shenzhen 518055, China; zhou-x20@mails.tsinghua.edu.cn

**Keywords:** multimorbidity, chronic diseases, elderly, bibliometric analysis

## Abstract

Objective: This study aimed to analyze the progression and trends of multimorbidity in the elderly in China and internationally from a bibliometric perspective, and compare their differences on hotspots and research fronts. Methods: Publications between January 2001 and August 2021 were retrieved from WOS and CNKI databases. Endnote 20 and VOSviewer 1.6.8 were used to summarize bibliometric features, including publication years, journals, and keywords, and the co-occurrence map of countries, institutions, and keywords was drawn. Results: 3857 research papers in English and 664 research papers in Chinese were included in this study. The development trends of multimorbidity in the elderly are fully synchronized in China and other countries. They were divided into germination period, development period, and prosperity period. Research literature in English was found to be mainly focused on public health, and the IF of the literature is high; In China, however, most research papers are in general medicine and geriatrics with fewer core journals. Co-occurrence analysis based on countries and institutions showed that the most productive areas were the United States, Canada, the United Kingdom, and Australia, while the Chinese researchers have made little contribution. The clustering analysis of high-frequency keywords in China and around the globe shows that the hotspots have shifted from individual multimorbidity to group multimorbidity management. Sorting out the top 10 highly cited articles and highly cited authors, Barnett, K’s article published in Lancet in 2012 is regarded as a milestone in the field. Conclusion: Multimorbidity in the elderly leads to more attention in the world. Although China lags behind global research the research fronts from disease-centered to patient-centered, and individual management to population management is consistent.

## 1. Introduction

With the changes in the disease spectrum and aging of the population, chronic diseases have become the most important disease burden and the main cause of death for the global population. It is estimated that chronic disease-related deaths accounted for 73.9% of the total deaths in 2019, ranking first among the causes of death [1]. Nowadays, more and more elderly people suffer from multiple chronic diseases at the same time because of the high prevalence and long course of chronic diseases. According to the definition of the American Geriatrics Association, multimorbidity in the elderly refers to the existence of two or more chronic medical conditions, including common chronic diseases, geriatric syndromes, and geriatric problems [2]. In the past two decades, the global prevalence of multimorbidity in the elderly has shown an upward trend despite the large differences between regions, among which the prevalence has reached from 57% to 81% in China [3,4,5].

As a prominent global public health problem, multimorbidity in the elderly has received considerable critical attention. Previous research compared its prevalence patterns in elderly people from different regions [6,7,8,9,10] and different age groups [11,12]. With the increase in elderly people having multimorbidity, the substantive research literature has been published on the burden of disease caused by comorbid conditions [13,14,15]. Additionally, geriatric multimorbidity usually leads to poor health, especially increasing disability and mortality in the elderly [16]. Research data shows that the global average number of years of disability (YLDs) has been maintained at 0.15 years for a long time, which is closely related to the rising multimorbidity rate in the elderly [1]. As huge research has been published on senile syndromes and common problems in the elderly, common chronic diseases and weakness [17,18,19], multiple medications [20,21,22], falls [23], malnutrition [24,25,26] gradually became hot research topics for scholars. In addition, multimorbidity further complicates disease treatment and rehabilitation and increases the risk for mental and physical complications, which leads to lower healthcare resources utilization [27]. Researchers also attempt to build standardized management models for elderly comorbid patients to measure the allocation of medical resources and construct a reasonable social medical security system [28]. Overall, the scope of research on multimorbidity in the elderly is gradually broadening and going in-depth, covering the whole process of chronic diseases from diagnosis to follow-up management, involving epidemiology, pharmacology, gerontology, and other disciplines. To date, there are a few studies that have investigated the development and trends of multimorbidity, while hotspots are still unclear in this field. Therefore, this paper aims to demonstrate the research trends and hotspots. Meanwhile, through a comparative analysis of the research literature in Chinese and English, the research gaps between China and the rest of the world were identified.

## 2. Methods

Bibliometric methods, are an integral part of research evaluation methodology within the scientific and applied fields and are used increasingly when studying various aspects of science. Each research paper has its own methodology and a huge literature has been published [29]. In this study, the Web of Science core database (WOS) and China National Knowledge Infrastructure database (CNKI) were searched systematically. These databases were chosen because they have the oldest and most comprehensive records of citation indexes as well as the comparability and stability of statistics obtained from these two different data sources are ideal [30]. After screening, 3857 articles in English and 664 articles in Chinese were retrieved. The process of literature screening is shown in Figure 1 and Figure 2.

### 2.1. Search Strategy

Data for this paper were acquired from the WOS and CNKI databases for the period from 2001 to 2021. The search was initiated on 21 July 2021 and contained all articles having the terms “Multimorbidity OR Chronic complex patients OR Multimorbid older people OR Multiple Chronic Conditions OR Multi-Morbidity OR Chronic medical morbidities OR Multimorbidity of chronic diseases OR Hypertensive diabetes patients OR Comorbid conditions OR Multimorbid patients OR Multimorbidity patterns OR Chronic conditions OR Prevalence of multimorbidity OR Multimorbid older-aged adults OR Multimorbidity Patients” in the article title, abstract and keywords. The subjects were limited to Geriatrics Gerontology, Health Care Sciences Services, Public, Environmental Occupational Health, and General Internal Medicine.

### 2.2. Inclusion and Exclusion Criteria

All the data from those databases, including papers’ information such as author names, titles, journals, keywords, institutional affiliations, citations, and abstracts were downloaded, while the CNKI database couldn’t provide citations. All of these data were converted to an xls. file (Microsoft Excel) to check data error. Then all the downloaded data were filtered by exclusions, which were: (1) expert consensus on related diseases, (2) non-elderly group of research subjects, (3) infectious diseases research, (4) not meeting the article’s definition of comorbidity, (5) the cases, and (6) incomplete key information.

## 3. Statistical Analysis

As the research literature in English and Chinese literature come from different databases, and the differences in analytic scope cut across both the topical distinctions visible in the citation analysis and the scope of academic cooperation, with consequences for defining key findings in the field [31], Thus, this review was organized into separate discussions for each database, in part to show the research gaps between China and the rest of the world [32]. In this study, Endnote 20 and VOSviewer were used to separately describe the basic characteristics of publications, countries, institutions, keywords, and citations. VOSviewer is a software with text mining and advanced visual analysis functions, which can be used to construct co-occurrence analysis [33,34]. Co-occurrence analysis aids to quantify the common information in various data, it revealing the content association and common relationships of the information [35]. The research type of co-occurrence analysis is wide, including co-country analysis, co-institution analysis, and co-keyword analysis. After searching the literature, the publishing time, journals, countries, institutions, and keywords of the literature were extracted using Endnote 20. Then the analysis was separated into three steps: (1) a descriptive statistical analysis on the growth pattern, number, year, institution, country, and core journals of the publications; (2) a co-occurrence analysis on the keywords by using VOSviewer, and then (3) a co-citation analysis to sort out the most influential papers and authors in this field.

## 4. Results

### 4.1. Publication Growth Pattern

The trend of publication is a simple and effective way to analyze the research hotspots in the field of elderly comorbid patients. As shown in Figure 3 and Figure 4, global attention to multimorbidity in the elderly has grown rapidly in the past two decades, and the development trends in China and abroad have been relatively synchronized. According to the number of publications per year, they can be roughly divided into the incubation period (2001–2010), the development period (2011–2014/2016), and the prosperity period (2016/2017 to present). Before 2010, combating infectious diseases was a hot topic in public health, and the population with chronic diseases was small which had not generated scholars’ interest. From 2001 to 2010, there were ≤100 articles in English and ≤20 articles in Chinese per year. As changes in the spectrum of diseases increased the number of multimorbidity in the elderly significantly, the rapid development of multiple chronic diseases research fields after 2010 is no surprise. During the period from 2011 to 2014, the publications were between 100–300 per year all around the world. In contrast, Chinese research was relatively lagging, and the development period was ushered in from 2011 to 2016, with nearly 40 articles per year. Since then, aging and advances in medical technology have increased survival years for the elderly and people with comorbid conditions, so multimorbidity in the elderly attracted more attention from researchers. Due to time limitations, the number of publications in 2021 cannot be counted for the whole year, but it will still show a steady growth in the future from the current trend.

### 4.2. Journal Analysis

In total, 3857 articles in English were published in 530 journals, among which the three most published journals are Value in Health (200 articles, 5.19%), BMJ Open (184 articles, 4.77%), and Journal of General Internal Medicine (145 articles, 3.76 %). From Table 1, it can be seen that the articles on elderly comorbid patients are quite diversified in their subject coverage. Although most are published in the field of public health and population health, some of them involve internal medicine, nursing, and other fields.

The top 10 Chinese journals for multimorbidity are listed in Table 2. All of those articles were recorded by more than 300 journals. Among them, “Chinese General Medicine” published the most papers (49 articles, 7.37%), which was the core of Peking University. Other journals have low publications with poor quality. Although the multimorbidity research literature is concentrated into a small number of core journals, and the journals are quite diversified in their topics, covering general medicine, psychiatry, geriatrics, and so on, by which they may be attributed to more than one subject area.

As it can be seen from the comparison above, six of 10 journals are published by the Britain, followed by the USA and Netherlands, however, articles written by Chinese researchers are rare, and Chinese journals have a low contribution in this field.

### 4.3. Contribution and Relationship of Countries and Institutes

To identify the distribution and cooperation of countries and institutions, a bibliometric map about research countries and institutions was drawn (Figure 5 and Figure 6). Similar to the journal analysis, the Britain and USA were by far the most productive regions in the field of multimorbidity, followed by Australia, Canada, Germany, and Spain, and close cooperative relationships have been formed between these countries. Among them, the United States, as the largest contributing country on multimorbidity in the elderly, has formed close cooperation respectively with Canada, Australia, and the United Kingdom; Europe owns the main research groups, in Germany and Spain. As for institutions, the most productive institutions were the University of Michigan, the University of Toronto, and the University of Sydney, with a stable triangular cooperative relationship. In Europe, there are two major research institutions in the United Kingdom, The University of Glasgow and Oxford University, that stay in close cooperation with the three institutions mentioned above. Apart from these academic institutions, The Eppendorf Hospital in Germany has contributed, but it is relatively independent with little cooperation. On the other hand, as the statistical method only analyzes the first author’s country, China does not appear in the figure because of the small number of research articles in the literature (less than 20) in English, but there are 142 articles involving Chinese researchers, who are scattered in research areas and cooperative groups.

### 4.4. Core Keyword Co-Occurrence Analysis

Over time, a knowledge map of keyword co-occurrence can reflect hot topics, and burst keywords (keywords that are cited frequently over a period of time) may indicate frontier topics [36]. VOSviewer was utilized to construct a knowledge map of co-occurring keywords. In the 3857 foreign articles in English (Figure 7, Table 3). Among them, “Multimorbidity” (748 times), and “Chronic disease” (418 times) were the most frequent words which are reasonable because they are the search terms. Except for them, other keywords actually reflect hotspots and topics that researchers are focusing on. “Prevalence” (591 times),” “Care” (412 times), and “Health-care” (329 times) are the other most frequent words, which show that prevalence investigation and elderly care are major research fields on the comorbidity of the elderly. It can be seen from Figure 7 that all keywords connect to others with different colors, so we divided the core keywords (minimum number of a term occurrence = 20) into four clusters by colors to analyze (Table 4). The red group mainly involves the care and management of chronic diseases, including patient self-management and follow-up. The green group is closely related to the red group, and even has cross-mixing; but the green group pays more attention to the life quality of patients, including disease burden, ADLs, and mental state. The blue group is the branch of the green group that is far from the red group, focusing on the physical health status and related risk factors, and even the relationship between multiple diseases. From the perspective of public health, the yellow group has a broader theme, and it investigates the population’s comorbidity patterns.

Moreover, burst keywords were used to analyze the frontier topics (Figure 8). Unlike the above, Figure 8 shows each keyword with an average publication year in the research field. The yellower the node color, the latest the keyword appears; the bluer the node color, the longer the keyword appears. Overall, color distribution in the figure shows a trend from blue-violet in the bottom right to yellow-green in the top left, which means the topics have gradually focused on chronic diseases management and healthcare in recent years.

In contrast, the top 20 high-frequency keywords in 664 articles in Chinese are sorted into Table 5. Excluded search terms, “Depression” (144 times),” “Diabetes” (79 times), and wake (44 times) were found to be the most three hot words. It was also found that Sun Xirong, Ji Jianlin, Jiang Qi and other author cooperative groups promote depression and diabetes comorbidity research to become a hotspot in this field. Different from other scholars around the world, Chinese researchers pay great attention to the quality of life, and disease burden, while studies on risk factors of patients with multimorbidity are similar to studies from other countries. Meanwhile, they also focus on common chronic diseases in the elderly. The low quality of the research literature in Chinese might be because of the considerable number of research and literature on common comorbidity patterns because the data are easy to obtain and analyze. It is worth mentioning that due to the lack of real and reliable population data in China, there are fewer studies on the prevalence of comorbid populations.

Similar co-keyword analysis for the research literature in Chinese, the keyword clustering analysis, and keyword average annual analysis (minimum number of a term occurrence = 10), were conducted (Figure 9 and Table 6). It can be seen that the keywords in the literature in Chinese are closely related, especially the red group and the green group. They both describe common comorbidity patterns and related chronic disease types and can be combined into one. The yellow group is related to general medicine and geriatrics, focusing on the whole body rather than specific diseases. From the perspective of the public health field, the blue group discusses the risk factors, management models, and community prevention for the elderly with multimorbidity. Comparing the keywords in the literature in Chinese and English, they are basically similar, ranging from individual comorbidity patterns to population comorbidity management, covering personal quality of life, disease burden, and public multimorbidity patterns and risk factors. However, there is still a considerable amount of research literature in Chinese focusing on diseases, which may be closely related to disease-centered medicine in China in the early years.

The burst keywords analysis, shown in Figure 10, also shows the shift in research hotspots from 2014 to 2019. The color in the picture changes from left blue-green to right light yellow, implying that the hotspot of elderly comorbidity in the research literature in Chinese has changed from specific disease orientation to multiple chronic diseases and complex conditions and health management gradually. Especially, in the past two years, Chinese researchers have done more explorations on multiple medications, elderly comprehensive evaluation, and management models for the elderly with comorbid diseases.

### 4.5. Co-Citation Reference Analysis

Co-citation of references means two (or more papers) are cited by one or more subsequent papers at the same time [37], which is vitally important to demonstrate the research front with higher details and accuracy [29]. To better understand the research literature’s influence and connection in the field, the top 10 most cited papers were identified and are listed in Table 5. However, the Web of Science core database is the only research target because of the database limitations. The papers cited more than 10 times were divided into five clusters (Figure 11 and Table 7). Among them, the article by Barnett, K published in Lancet in 2012 was the most cited so far (478 citations), and it has a milestone significance in geriatric multimorbidity research progress. Most of the highly cited papers in this field involve research on epidemiology, including the prevalence rate of multimorbidity and common patterns of multimorbidity in different populations. These papers are highly cited in the “background introduction” part of other papers. Another part of the highly cited literature pays attention to primary medical resources utilization, trying to provide new ideas to manage the multiple chronic diseases patients in primary care.

## 5. Discussion

The bibliometric analysis provides an idea about the progress of research and contributions of the researchers in a specified field [38]. WOS and CNKI databases are both ideal platforms for bibliometric analysis, and it’s credible to compare the two databases because their text and citation information for all the articles was sufficient and perfectly matching [32]. In this study, two research databases were formulated to separately analyze the current situation and development trend of multimorbidity research from 2001 to 2021 in China and abroad, identify the contributions of different nations and institutions and their partnerships, and especially the role of Chinese scholars. Meanwhile, research hotspots and research front were tracked through cluster keyword mapping and burst terms analysis. At last, co-citation analysis was utilized in the WOS database, and the global trends and most contributing papers in multimorbidity in the elderly English and Chinese databases in this field were compared to encourage more dialogue among scholars [31]. By comparing the research differences on topics, trends, and hotspots, he aim was also to highlight the remaining gaps in China and the rest of the world to reveal opportunities for future research.

### 5.1. Research Status and Trend

As for publication years, the trend of publication is basically the same for the research literature in English and Chinese (Figure 3 and Figure 4). As the concept of senile multimorbidity has broadened, there exists a wide range of in-depth research. In the past five years, global publications in multimorbidity have increased year by year, indicating this field has gradually received attention from scholars, and the growth trend is expected to continue. For journals, the most contributive journals on multimorbidity are mainly about public health and population health, while some involve nursing, internal medicine, and other fields, which means multimorbidity has different emphases and involves the intersection of many fields (Table 1 and Table 2). Compared with the English journals, the Chinese journals have little influence (both in quantity and quality). Instead of public health, general medicine, spiritual and psychological studies are also substantive in China. As for the future development in this field, it may still be showcased in Value in Health and BMJ open journals since they published the most articles in the past. In addition, the USA is the leading country for research in multimorbidity, followed by the U.K and Australia. There exists a stable cooperative relationship between North America and Europe. The institutions listed in Figure 6 could potentially become partners for cooperation among those who are committed to this field. With lots of funding guarantees in this field in China, top Chinese institutions are expected to emerge in the future.

On the other hand, co-citation analysis was utilized and it showed that the article by Barnett, K published in Lancet in 2012 was the most cited so far (Table 7), which greatly promoted the development of this research field. As shown in Figure 11, the most highly co-cited literature was based on epidemiological basic surveys (e.g., prevalence, co-disease types) to lay a solid foundation for the treatment, care, and management of later co-patient populations.

### 5.2. Research Hotspots and Topics

Judging from the analysis of research hotspots, global hot research topics mainly revolve around focuses on the epidemiological prevalence and individual health management, and then gradually expand to elderly care, daycare, and quality of life. However, keyword analysis in China shows that comorbidity of physical and mental disorders is still the main research direction of multimorbidity, and the common multimorbidity includes diabetes, hypertension, depression, and anxiety. At the same time, some research uses public databases such as CHARLS and CFPS to conduct epidemiological analysis on the prevalence of multimorbidity and quality of life of the population with high IF.

The network map indicated that the Chinese research trends are slightly behind research literature in English (Figure 7 and Figure 9). At present, the trends on multimorbidity in the elderly in the research literature in English have gradually shifted from individuals to populations at the intersection of geriatrics and public health. Although care and management is still the main problem in the field which is still necessary to further strengthen the classification of elderly multimorbidity and explore scientific and efficient management models. There were also many breakthroughs in patients’ quality of life, disease burden, and comorbidity management models [39,40,41]. In contrast, Chinese research mostly focuses on specific chronic diseases or aging issues. Among them, physical and mental disorders are the most selected research fields by domestic researchers, and a large number of studies on depression, anxiety with insomnia, hypertension, diabetes, and other types of elderly multimorbidity have been published [42,43]. It is worth mentioning that a research team devoted to the comorbidity pattern of depression and type 2 diabetes has been formed in China, and team members have reached close cooperation in this field. However, some Chinese research literature that refers to the hotspots in the research literature in English to conduct epidemiological investigations on the prevalence patterns and prevalence of multimorbidity, and try to build models to measure the quality of life and disease burden. This type of literature is mostly published in core journals. Generally, it can be seen that the IF of Chinese research literature is not high, due to its low research innovation, a small research population, and simple research methods.

### 5.3. Research Fronts

Currently, it can be safely forecasted that future research works in English will continue to tilt towards public health, and gradually transit from the management of comorbid populations to comorbidity prevention and health control. Chinese research will go along with the development trends of other countries and will transit to epidemiological investigations and community management models for the comorbid populations. In terms of the epidemiological investigation of comorbidity in the elderly, Hu Xiaolan conducted a systematic review of the Chinese research on the prevalence of multimorbidity in the elderly in 2015 and found that there is an obvious heterogeneity in the survey methodology [44], which proves that there are big differences in sample selection, statistical caliber, and definition of comorbid diseases in the existing studies in China. In the future, the standardized guidance of research should be strengthened to enhance the management prevention, and control of comorbid diseases. Meanwhile, the management of the elderly comorbid populations and the establishment of a large cohort of the elderly comorbid populations for interventional research will become new hotspots. Although the current domestic and global cooperation is not relatively close, with the convergence of future research trends and strong support from Chinese funds, more Chinese researchers are expected to have scientific research output in this field.

### 5.4. Limitations

Due to the database limitations, the co-cited research literature was not analyzed. So high-quality Chinese articles that have made important contributions in this field cannot be sorted out and cannot be compared with the articles in English. In addition, this paper only selected two databases to represent the research status in China and abroad and does not consider the publication bias of the literature. In the future, relevant research can further expand the scope of literature retrieval and analyze current research hotspots and trends more comprehensively.

## 6. Conclusions

Through the analysis and comparison of research literature databases in English and Chinese, it can be seen that multimorbidity in the elderly has rapidly become a research hotspot in the world. Although China lags behind global research, the development trend from disease-centered to patient-centered, from individual management of multimorbidity to population management is consistent.

## Figures and Tables

**Figure 1 ijerph-19-00353-f001:**
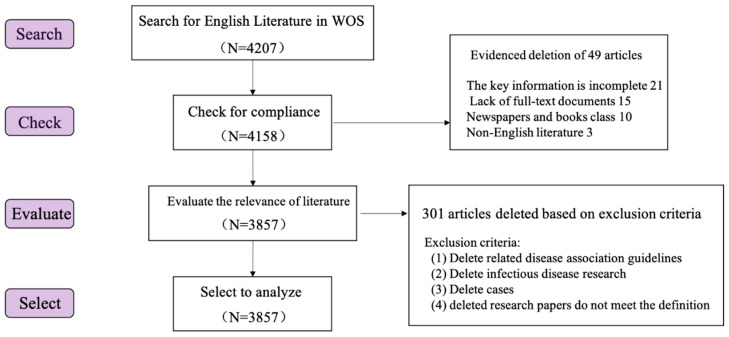
Global literature retrieval process.

**Figure 2 ijerph-19-00353-f002:**
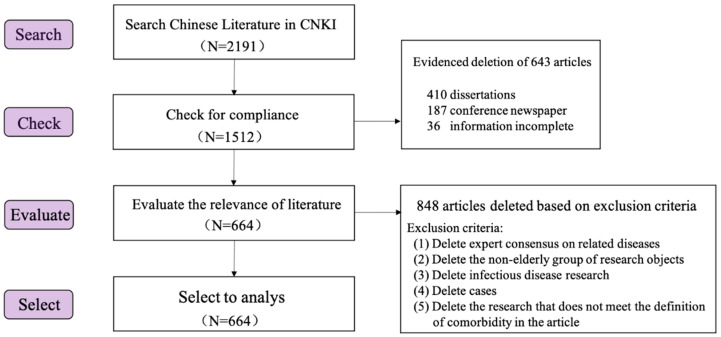
Domestic document retrieval process.

**Figure 3 ijerph-19-00353-f003:**
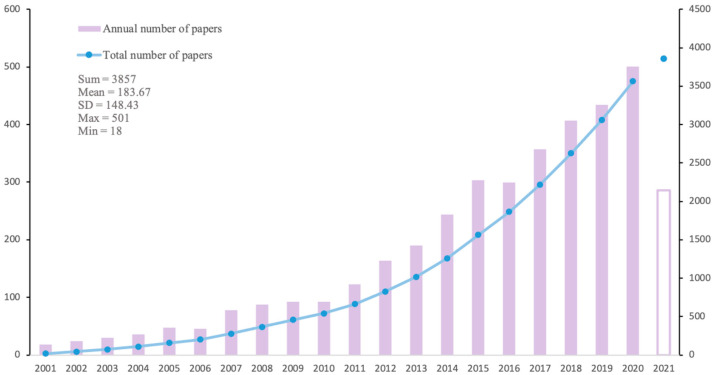
Trends of global research publications.

**Figure 4 ijerph-19-00353-f004:**
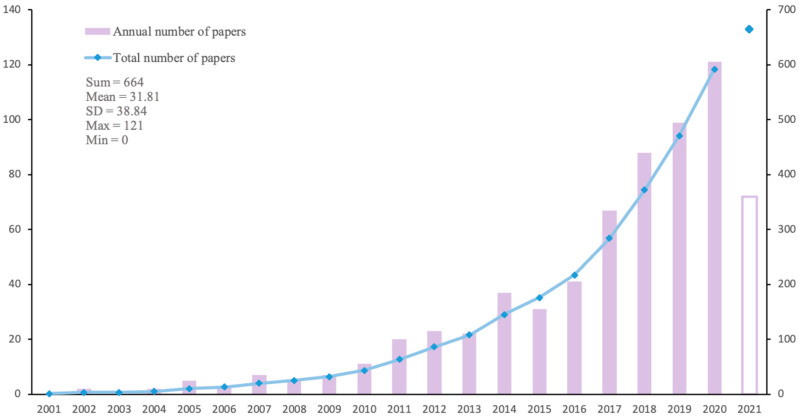
Trends in domestic research and publication.

**Figure 5 ijerph-19-00353-f005:**
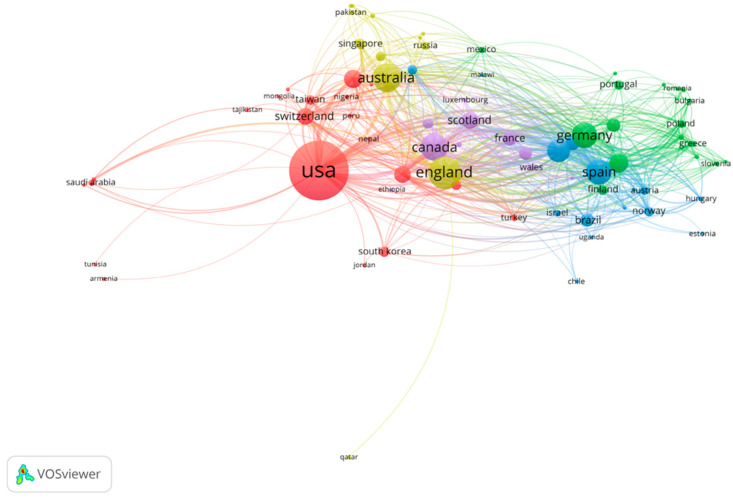
Co-countries analysis of global research.

**Figure 6 ijerph-19-00353-f006:**
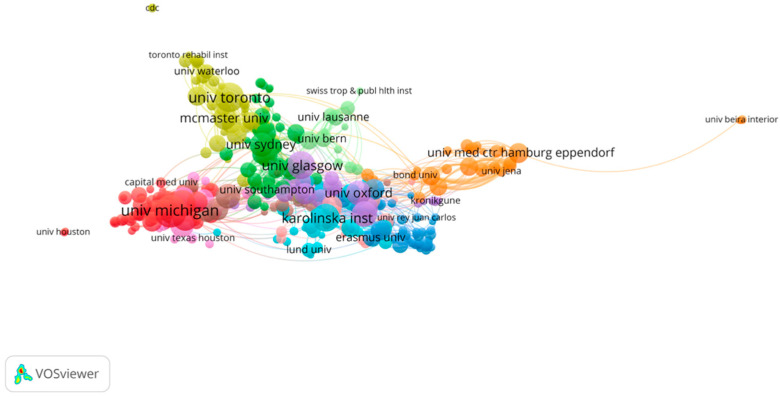
Co-institution analysis of global research.

**Figure 7 ijerph-19-00353-f007:**
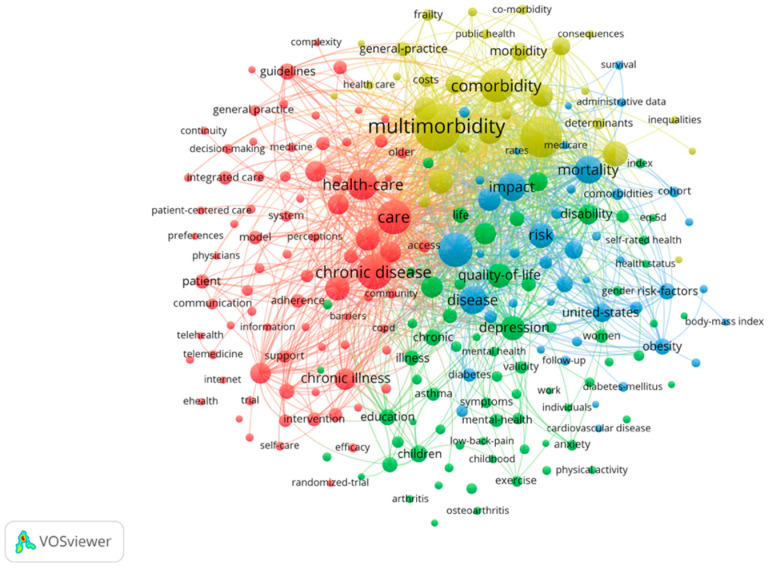
Co-keywords analysis of global high-frequency.

**Figure 8 ijerph-19-00353-f008:**
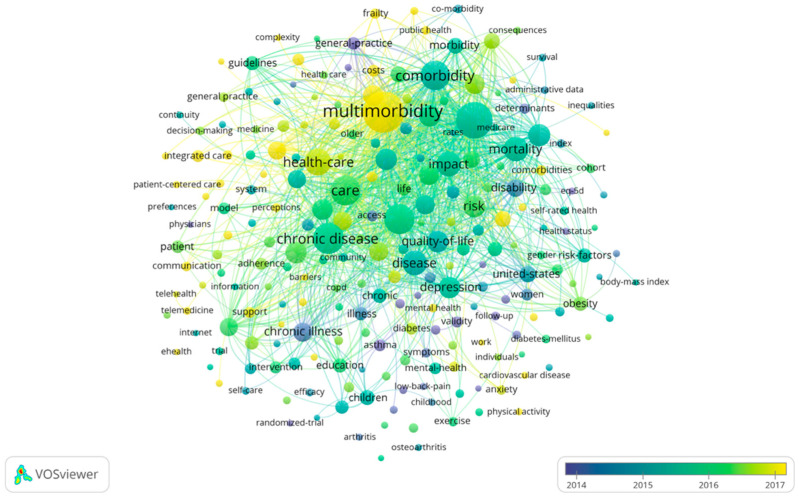
Burst keywords analysis of global high-frequency.

**Figure 9 ijerph-19-00353-f009:**
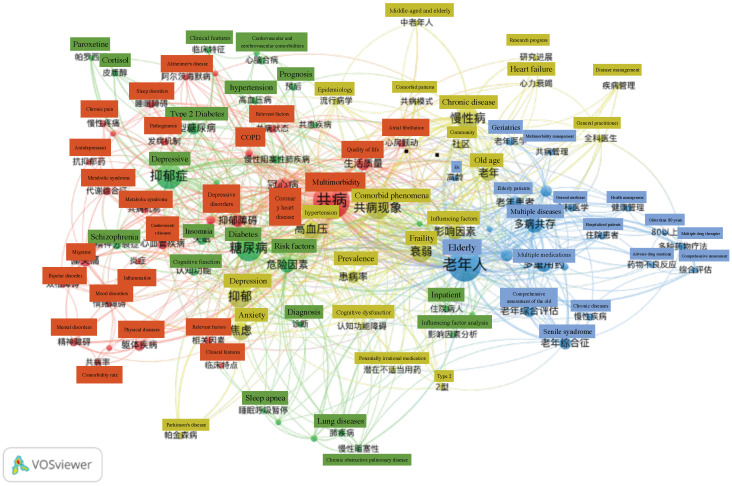
Co-keywords analysis of Chinese high-frequency.

**Figure 10 ijerph-19-00353-f010:**
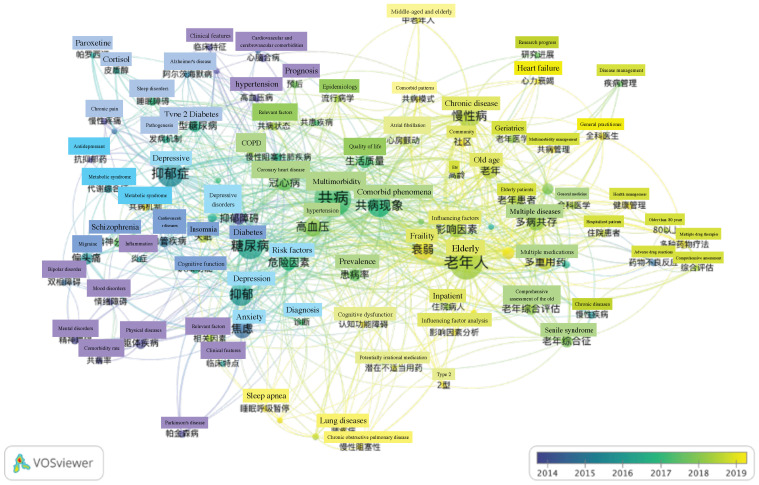
Burst keywords analysis of Chinese high-frequency.

**Figure 11 ijerph-19-00353-f011:**
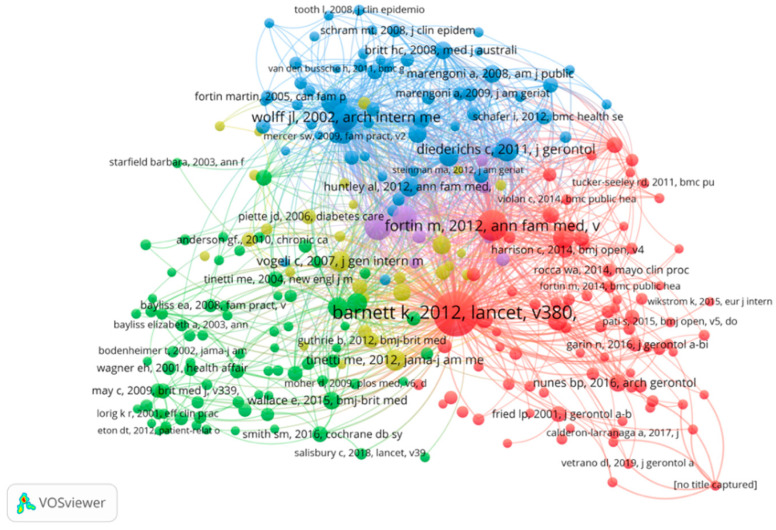
Co-cited analysis of global publications.

**Table 1 ijerph-19-00353-t001:** Top 10 English Journals Contributing.

TOP	Journal Title	Country	Number	IF
**1**	Value in Health	America	200	5.725
**2**	BMJ Open	Britain	184	2.692
**3**	Journal of General Internal Medicine	America	145	5.128
**4**	Quality of Life Research	Netherlands	101	2.773
**5**	International Journal of Integrated Care	Netherlands	100	2.753
**6**	European Journal of Public Health	Britain	86	2.391
**7**	BMC Public Health	Britain	83	2.837
**8**	BMC Health Services Research	Britain	83	2.193
**9**	BMC Family Practice	Britain	71	2.290
**10**	International Journal of Environmental and Health Research	Britain	65	1.916

**Table 2 ijerph-19-00353-t002:** Top 10 Chinese journals Contributing.

TOP	Journal Title	Number	IF	The Core or Not
**1**	Chinese General Practice	49	2.25	Core
**2**	Chinese Journal of Multiple Organ Diseases in the Elderly	19	1.043	Not
**3**	Chinese Journal of Clinical Healthcare	15	0.987	Not
**4**	Journal of Clinical Psychiatry	15	0.99	Not
**5**	Chinese Journal of Geriatrics	14	1.049	Core
**6**	Modern preventive medicine	13	1.801	Core
**7**	International Journal of Psychiatry	12	1.041	Not
**8**	Chinese Journal of Gerontology	12	1.187	Core
**9**	Digest of the world’s latest medical information	9	0.514	Not
**10**	Practical geriatrics	9	0.756	Not

**Table 3 ijerph-19-00353-t003:** Top 20 Frequency of Core Keyword in English Articles.

TOP	Key Words	Number	TOP	Key Words	Number
**1**	Multimorbidity	748	**11**	Population	217
**2**	Prevalence	591	**12**	Quality-of-life	213
**3**	Chronic disease	418	**13**	Management	211
**4**	Care	412	**14**	Chronic diseases	208
**5**	Health	403	**15**	Depression	208
**6**	Health-care	329	**16**	Primary-care	202
**7**	Disease	280	**17**	Multiple chronic conditions	193
**8**	Mortality	277	**18**	Outcomes	188
**9**	Impact	265	**19**	Adults	182
**10**	Risk	226	**20**	Epidemiology	174

**Table 4 ijerph-19-00353-t004:** Cluster keywords analysis of English articles.

Cluster Name	Selected Terms for Each Cluster	Legend	Number of Clusters
**#1 Chronic disease care and management**	Chronic disease (418), Care (412),Health-care (329), Management (211),Outcomes (188), Chronic illness (162),Quality (153), Self-management (146)	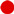	81
**#2 Patient quality of life/status**	Quality-of-life (213), Depression (208),Chronic Conditions (166), Burden (128),Older-Adults (164), Disability (144),	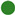	69
**#3 Research on health status and risk factors**	Health (403), Disease (280), Mortality (277),Impact (265), Risk (226), Adults (182)	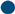	44
**#4 comorbidity model**	Multimorbidity (748), Comorbidity (388), Multiple chronic conditions (193),Population (217), Primary-care (202) Epidemiology (174)	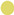	30

**Table 5 ijerph-19-00353-t005:** Top 20 Frequency of Core Keyword in Chinese Articles.

TOP	Key Words	Number	TOP	Key Words	Number
**1**	Comorbidity	152	**11**	Risk factors	29
**2**	depression	144	**12**	Multi-drug	27
**3**	The elderly	126	**13**	Influencing factors	26
**4**	diabetes	79	**14**	Coronary Heart Disease	26
**5**	chronic	61	**15**	Elderly patients	22
**6**	Comorbidity	58	**16**	Quality of Life	21
**7**	weak	44	**17**	Prevalence	20
**8**	anxiety	39	**18**	Old age syndrome	17
**9**	hypertension	37	**19**	disability	15
**10**	Multiple diseases coexist	32	**20**	management	14

**Table 6 ijerph-19-00353-t006:** Cluster keywords analysis of Chinese articles.

Cluster Name	Selected Terms for Each Cluster	Legend	Number of Clusters
**#1 main types of multimorbidity**	Multimorbidity (152), quality of life (21), depressive disorder (19), physical disease (14), cardiovascular disease (14), metabolic syndrome (8), depression (64), anxiety (39)	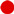	28
**#2 related chronic diseases**	Diabetes (79), depression (80), hypertension (37), risk factors (29), coronary heart disease (26), type 2 diabetes (18), migraine (18), schizophrenia (10), recognition Knowledge function (10),	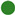	27
**#3 Geriatrics**	The elderly (129), coexistence of multiple diseases (32), multiple medications (27), elderly patients (22), comprehensive assessment of the elderly (18), senile syndromes (17), general medicine (9)	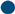	21
**#4 Comorbidity patterns and related factors**	Chronic diseases (61), multimorbidity (58), frailty (44), influencing factors (26), prevalence (20), community (10)	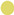	21

**Table 7 ijerph-19-00353-t007:** TOP 10 highly cited papers in English.

TOP	Title	Year	Number of Cited	Journal
**1**	Epidemiology of multimorbidity and implications for health care, research, and medical education: a cross-sectional study	2012	478	Lancet
**2**	A Systematic Review of Prevalence Studies on Multimorbidity: Towards a More Uniform Methodology	2012	226	Annals of Family Medicine
**3**	Prevalence, expenditures, and complications of multiple chronic conditions in the elderly	2002	167	Archives of Internal Medicine
**4**	Epidemiology and impact of multimorbidity in primary care: a retrospective cohort study	2011	165	British Journal of General Practice
**5**	Prevalence of multimorbidity among adults seen in family practice	2005	158	Annals of Family
**6**	The prevalence of multimorbidity in primary care and its effect on health care utilization and cost	2011	138	Family Practice
**7**	Methodological challenges concerning the selection of diseases for a standardized multimorbidity index	2011	132	BUNDESGESUNDHEITSBLATT-GESUNDHEITSFORSCHUNG-GESUNDHEITSSCHUTZ
**8**	Multiple chronic conditions: Prevalence, health consequences, and implications for quality, care management, and costs	2007	106	Journal of General Internal Medicine
**9**	Designing Health Care for the Most Common Chronic Condition-Multimorbidity	2012	102	Jama-Journal of the American Medical Association
**10**	Multimorbidity in Older Adults	2013	102	Epidemiologic Reviews

## Data Availability

Exclude this statement.

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
