# Peer review of "Multimorbidity in the Elderly: A Systematic Bibliometric Analysis of Research Output"

_ijerph, 2021, doi:10.3390/ijerph19010353_

Round 1

Reviewer 1 Report

The work presents a good organization of a systematic review, to which perhaps to put some obstacle, perhaps the review has been limited to few scientific search engines, which could possibly have given them another perspective in the review, otherwise the biometric analysis of the results is novel in the publications of systematic reviews, so it provides new ideas to continue with this type of review process.

Some tables and graphs at the beginning of the text may be excessively larger and longer than normal with a font larger than the text. For example table 2 and 10.

Reviewer 2 Report

Dear authors,

Since you're using the Chinese database for analysis, I think you'd have to submit it to a Chinese academic journal.

methods → Why is the search criterion in Web of Science different from those in the Chinese database? I can't understand criterion of consort diagrams, because of insufficient presentation. It was difficult for me to understand the method in the first place and the subsequent conclusions.  

Reviewer 3 Report

I think it is confusing and inaccurate to call this paper a 'systematic review'. It is a bibliometric analysis.

Readers will be interested in the paper from the perspective of the methods used and the novel mapping techniques: these do yield interesting findings about the topic area of multimorbidity in the elderly. Some readers may be a little overwhelmed by all the metrics: I did wonder whether all were needed.

The paper needs editing for English. Below I list some of the more obvious infelicities:

p. 1, line 29 - global people - change to global population

p. 1, line 40 - the prevalence of multimorbidity and their prevalence patterns - change to its prevalence patterns

p. 1, line 44 - usually lead -- change to usually leads

p. 4, line 123 - in contract - should be in contrast

p. 7, line 177 - a great contribute - should be contribution

p. 14, line 269 - co-cited reference analyze - should be analysis

p. 17, line 234 - last but not list - should be last but not least

p. 18, line 378 - founders: no founding - should be funders: no funding

The authors’ aim seems to be to chart the development of the field of multimorbidity in the elderly using bibliographic metrics and to present findings from this analysis. The literature examined is that captured by WOS (global) and that in Chinese databases. 

This approach is certainly not original. Boyack KW & KLavans R wrote about ‘co-citation analysis, bibliographic coupling, and direct citation’ in the J.Am.Soc.Inf.Sci.Technol. in 2010. Aliya Saperstein, Andrew M Penner, and Ryan Light used these bibliographic methods in ‘Racial formation in perspective: Connecting individuals, institutions, and power relations’ (Amer. Rev. Sociol. 2013. 39:359-78). Perhaps these should have been cited. The application of these methods to multimorbidity in the elderly is new. 

The contribution of these methods to multimorbidity in the elderly is new, including the comparison of the field in China with that globally. This narrative history approach can yield novel findings. 

I think the methodology is OK. I do think it rather dominates the paper and may put off some readers who don’t particularly like these “techie” approaches. Maybe more emphasis could be placed on interpretation to pull in readers. 

As I have said they don’t really pose questions or hypotheses. 

They could add a few more references on both the methods and applications, including those cited above. 

The tables and figures are fine. I do think the paper has merit as, apart from Saperstein et al., there are few published papers that have applied these methods to a particular area of study. However, I think the authors should guard against the bibliographical analysis (as shown in figures and table) dominating the paper. It is a tool of analysis and not the main story! 

Round 2

Reviewer 2 Report

Dear authors,

Thanks for your revising the manuscript.

I think it's a good idea to publish in the current form.

Have a wonderful end to the year.